# Incidence and predictors of anemia among adults on HIV care at South Gondar Zone Public General Hospital Northwest Ethiopia, 2020; retrospective cohort study

**Agimasie Tigabu** [1] *, **Yeshiwork Beyene**[2], **Temesgen Getaneh**[3], **Bogale Chekole**[4], **Tigist Gebremaryam**[5], **Ermias Sisay Chanie**[6], **Nigusie Selomom** [6], **Tamiru Alene**[7], **Getachew Aragie**[6], **Getasew Legas** [8], **Getnet Dessie**[2]

1 Lecturer of Nursing, Department of Adult Health Nursing, College of Medicine and Health Sciences, Debre Tabor University, Debre Tabor, Northwest Ethiopia, 2 Department of Adult Health Nursing, School of Health Science, College of Medicine and Health Sciences, Bahir Dar University, Bahir Dar, Northwest Ethiopia, 3 Department of Clinical Midwife, College of Medicine and Health Sciences, Debre Markos University, Debre Markos, Ethiopia, 4 Department of Pediatrics Nursing, College of Medicine and Health Sciences, Wolkite University, Wolkite, Ethiopia, 5 Department of Pediatrics Nursing, College of Medicine and Health Sciences, Debre Markos University, Debre Markos, Ethiopia, 6 Department of Pediatrics Nursing, College of Medicine and Health Sciences, Debre Tabor University, Debre Tabor, Ethiopia, 7 Department of Pediatrics Nursing, College of Medicine and Health Sciences, Wolaita Sodo University, Sodo, Ethiopia, 8 Department of Psychiatry Nursing College of Medicine and Health Sciences, Debre Tabor University, Debre Tabor, Ethiopia

* ethiomom23@gmail.com

**Data Availability Statement:** All relevant data are within the manuscript and its Supporting Information files.

# Abstract

## Background

Anemia is a major public health problem worldwide which accounts 24.8% of the population. Subsequently, anemia is a leading killer of people living with human immunodeficiency virus and many of these deaths occur in developing countries including Ethiopia. Cross sectional studies have done on anemia and human immunodeficiency virus. However, there is limited study on incidence of anemia and its predictors among adults on HIV care, especially no survival study has been conducted in the study area.

## Objective

To assess incidence and predictors of anemia among adults on Human immunodeficiency virus care.

## Methods

An institution-based retrospective cohort study was conducted among 434 adults on HIV care from January 1st 2015 to December 30th 2019 at Debre Tabor Referral Hospital. A computer-generated simple random sampling technique was employed to select the study participants. Ethical clearance was obtained from the Institutional Review Board of Bahir Dar University, and also, we got implied consent to review charts from the concerned bodies in the hospital. Data were entered using Epi-data version 3.1 and analyzed by using STATA

**Funding:** The author(s) received no specific funding for this work.

**Competing interests:** The authors have declared that no competing interests exist.

**Abbreviations:** AIDS, Acquired Immunodeficiency Syndrome; AHR, Adjusted Hazard Ratio; ALT, Alanine Amino Transferase; ART, Antiretroviral Therapy; AZT, Zidovudine; BMI, Body Mass Index; CD4, Cluster of differentiation 4; CPT DTG, Cotrimoxazole Prophylactic Therapy; DTGH, Dolutegravir Debre Tabor General Hospital; FMOH, Federal Ministry of Health; FDC, Fixed-Dose Combination; HAART, Highly Active Antiretroviral Therapy; HIV, Human Immunodeficiency Virus; HGB, Hemoglobin; OI, Opportunistic Infection; PLWHA, People Living With HIV/AIDS; RBC, Red Blood Cell.

version 14.0. A Kaplan Meier survival curve was utilized to estimate anemia free survival time. Bivariable and Multivariable Cox proportional hazards model were fitted to identify predictors of anemia.

## Results

The overall incidence density rate of anemia was 6.27 (95% CI: 0.051, 0.077) per 100 person years. Clinical stage III/IV (AHR = 1.04; 95% CI = 1.02, 1.06), Body Mass Index less than 18.5 kg/m2 (AHR = 3.11; 95% CI = 1.56, 6.22), serum creatinine greater than 1.1 IU/L (AHR = 2.07; 95% CI = 1.12, 3.81) and fair/poor level of adherence(AHR = 1.05; 95% CI = 1.03, 1.07) were statistically significant predictors of anemia while increased anti-retroviral treatment duration (AHR = 0.98; 95% CI = 0.97, 0.99) decrease the risk of anemia at 95% confidence level.

## Conclusion

The overall incidence density rate of anemia was high. Patients with clinical stage III/IV, body mass index < 18.5 kg/m$^2$, serum creatinine greater than 1.1 IU/L and fair/poor level of adherence were significant predictors of anemia while increased antiretroviral treatment duration had decreased the risk of anemia.

## Recommendation

Even if the overall incidence rate of anemia was lower as compared to previous studies in Ethiopia, still the incidence of anemia was high. So, prevention measures should be taken beside with HIV care especially within 6-months ART initiation.

## Background

For human immunodeficiency virus [HIV] infected nonpregnant women and men whose age ≥15 years old, anemia defined as; the level of hemoglobin concentration < 11 g/dl [1–3].

The causes of HIV-related anemia include impaired hematopoietic due to increased cytokine production, decreased endogenous erythropoietin production, blood loss, hemolytic that may result from red blood cell [RBC] autoantibodies, infiltration of the bone marrow by neoplasm or HIV infection itself, opportunistic infections with Mycobacterium avium complex or parvovirus B-19 and drugs such as zidovudine or cotrimoxazole [4–7].

Anemia is a major public health problem that affects an estimated 1.62 billion people worldwide which equivalent to 24.8% the population [8, 9], in which, one of the sufferers are HIV infected individuals. The prevalence of anemia among HIV positive adults ranges between 16.2% and 69% in Africa [10–17] and between 11.4% and 42.9% in Ethiopia [12, 18–20].

Aside from the total diseases occurrence patients with symptomatic HIV infection account for 75–80% of comorbidity [21]. Regarding severity of the diseases, it increases with a decrease in Cluster of differentiation 4 [CD4] cell counts and advancement of HIV infection [3, 22, 23].

The risk of acquired immunodeficiency syndrome [AIDS] related death was more than double for patients with anemia compared with those without anemia due to its association with shorter time to immunologic disease progression, greater need for transfusion and poor quality of life [24–26].

Anemia among HIV patients can lead to impaired physical functioning, psychological distress, poor quality of life, accelerated disease progression and shorter life expectancy [27, 28] patients also suffer from fatigue, weakness, dyspnea, pallor, lethargy, depression, and impaired cognitive function [3, 24, 29].

The Potential factors for HIV patients at risk for developing anemia were female sex, patients with low CD4 cell count < 200 cells per mm3 and patients with advanced disease stage III / IV [30–32] and may result from genetic disorders, chronic diseases, and nutritional deficiencies [33, 34].

Since test and treat programs launched in 2013, it has achieved remarkable improvements in the prevention of HIV related opportunistic infections [OIs] [35–37]. Ethiopia is one of the countries, which adopted and implemented this program for HIV positive adults since June 2017 [38].

Optimization of current antiretroviral therapy(ART) regimens is a critical component to support country efforts to achieve the goal of end of acquired immunodeficiency syndrome [AIDS] pandemic as a public health threat by 2030 [39]. Following this, more than 50 low- and middle income countries (LMICs) are including or planning to include dolutegravir (DTG) containing regimens in their national protocols, as the preferred first-line option, particularly the fixed-dose combination (FDC) tenofovir/lamivudine/dolutegravir at the end of 2017. Even though Ethiopia is one of the countries which adopted and implemented this new HIV drug in 2018 [40], its impact on anemia is not understood.

Given the importance of knowing the incidence of anemia and its predictors in low-income settings is critical for improving health of HIV positive patients. However, research on the incidence of anemia in low-income countries is relatively sparse.

Few available cross-sectional studies cannot address the incidence and predictors of anemia after the new regimen has been started especially, the study area. To address this gap in the literature, this study aims to assess the incidence and predictors of anemia among HIV-infected adults at Debre Tabor general hospital.

## Methods

### Study setting and period

The study was conducted at Debre Tabor General Hospital (DTGH) ART clinic from January 1, 2015, to December 30, 2019. Debre Tabor General Hospital is found in Debre Tabor town. Debre Tabor is the capital town of the south Gondar zone. It is located 667 kilometers northwest of Addis Ababa, which is the capital city of Ethiopia and about 99 kilometers northeast of Bahir Dar, the capital city of the Amhara region. The Hospital is the only general hospital in South Gondar Zone which started giving ART service in 1997 E.C and the case-team comprised trained physician, nurses, pharmacists, laboratory technicians, data clerks, and ART education adherence counselors.

### Study design and population

An institutional-based retrospective follow-up study was conducted at Debre Tabor General Hospital ART clinic. All HIV positive adults ever started ART at DTGH ART clinic who have follow up from January 1, 2015, to December 30, 2019 GC, and whose card available at the time of data collection was study population. Non-pregnant women and men HIV-infected adults and adolescents whose age ≥15 years old were included in the study. Adults who had anemia at the beginning of the follow-up and incomplete baseline data were excluded from the study.

## Operational definitions

**Event.**   The occurrence of anemia for HIV-positive adults after ART initiation until the end of the study that ascertained by patient document review.

**Censored.**   Adults did not develop anemia (transfer out to other services, switch off antiretroviral therapy, Death, drop out, and still on ART in the Hospital) that ascertained by patient document review.

*Level of adherence.* Estimate adherence level using the table below taken from ART intake form.

Adherence percent missed doses.

| Level of adherence | Percent (%) | (of 30 doses) | (of 60 doses) |
|---|---|---|---|
| **G(good)** | > 95% | < 2 *doses* | < 3 *doses* |
| **F(fair)** | 85–94% | 2–5 doses | 3–9 doses |
| **P(poor)** | < 85% | ≥ 6 *doses* | > 9 *doses* |

## Sample size and sampling procedure

The sample size was determined by a two-sample comparison of survival functions Log-rank test via Stata version 14.0 using 5% significant level, 80% power and one to one non exposed to exposed ratio, and considering significant predictors of developing anemia in the previous study in Ethiopia [41]. The final sample size was 434. The total records in the hospital among eligible study population were 2157 which was taken from Excel data in the hospital. 434 Charts were retrieved based on their medical registration number which was selected using a computer generated simple random sampling technique.

## Data collection instrument and quality control

The data extraction checklist was adapted from the Federal Ministry of Health ART follow-up forms. Data were collected using this validated and reliable checklist. Patient's socio-demographic variables, Clinical and treatment-related variables of HIV infected Adults were extracted from their chart. Pretest was conducted among 44 medical records to check the consistency of the abstraction tool. Two-day training was provided for data collectors and supervisor about how to review the documents, extract data from medical records for data collectors and about the entire data collection process. The filled formats were checked for completeness by the data collectors, supervisor and finally by principal investigator on daily bases.

## Data collection

Data were collected by three BSc Nurses that had comprehensive HIV care training. An additional health care provider who has ART training certificate has participated as a supervisor. Once data extraction from patient charts has been completed, code was given for each chart to avoid duplication. The data collection period was from January 1, 2015, to December 30, 2019 GC.

## Data processing and analysis

Clear and completed data were entered using EPI-data Version 3.1 and were analyzed using STATA Version 14 statistical software. Descriptive statistics were summarized using percentage and median, and presented using tables and figures. At the end of follow-up, the outcome of each study participant was dichotomized into censored or event. Assumption of Cox

proportional hazard regression model was checked using Schoenfeld residual and Log-Log plot tests. In addition, the model goodness of fit was assessed using Cox-Snell residual test. The Kaplan Meier survival curve was used to estimate the anemia free survival time of HIV-positive adults on ART. Log-rank test was used to compare the survival curves of different categorical explanatory variables. Bi-variable Cox-proportional hazard regression model was used to select eligible variables for the final model. Variables having $p$-value ≤0.25 in the bi-variable analysis were fitted into the multivariable Cox proportional hazard regression model. Finally, adjusted hazard ratio with its corresponding 95% confidence interval was conveyed to declare the presence of significant association between the predictor and outcome variables.

### Ethics approval and permission to chart review

Ethical clearance was obtained from Ethical Review Board of Bahir Dar University. Permission letter was obtained from concerned bodies of Debre Tabor General Hospital to review charts. Names and unique ART numbers of patients was not included in the checklist. Moreover, data collectors and the supervisor were health professionals who have work experience in the ART clinic to maintain confidentiality of people living with HIV/AIDS. Information retrieved was used only for the study purpose.

## Results

### Socio-demographic characteristics of adults on ART

A total of 434 chart of HIV positive adults whose age ≥ 15 years old were reviewed. Of these, 411 HIV positive patient's medical records were included in the analysis with completeness rate of 94.7%. The median age of the entire cohort was 34 years (IQR; 12). More than half of religion follower were Orthodox Christian 323(78.6%) followed by Muslim 60(14.6%) and protestant 28(6.8%) follower and other Socio-demographic variables are showed in **Table 1** below.

### Clinical and treatment related characteristics

The mean ART duration of the entire cohort was 86.75 months (IRR: 81.6–92). Two hundred six (60.1%) patients had changed their initial regimen during the follow up period. From the patients who have changed their initial regimen, 26(12.6%) patients switched to second-line HAART. The reason for changing the initial regimen was due to new drug available 99(48%) and due to drug side effects 80(40.2%). However, the remaining reason for changing initial regimen was not recorded (**Table 2**).

### Incidence of anemia

Four hundred eleven (411) study participants were followed for five years which gave us 1419.63 person years of observation. During the follow up period, 89 new anemia cases were observed.

Hence, the overall anemia incidence density rate (IDR) in the cohort was 6.27 (95% CI: 0.051, 0.077) per 100 person years of observation and cumulative incidence was 21.65% while 322 (78.35%) were censored. Of the censored patients, 262 (81.4%) didn't develop anemia until the end of the study, 31 (9.6%) were transferred out, 18 (5.59%) were dropped out, 4 (1.24%) were lost to follow up and the remaining 7 (2.17%) have died.

Study participants were followed for a minimum of 0.67 and a maximum of 60.63 months. The median follow-up period was 49.03(IQR; 35.3) months.

**Table 1. Baseline socio demographic characteristics of adults on HIV care.**

| Characteristics | Number | Percent |
|---|---|---|
| **Age** | | |
| 15 to <30 | 134 | 32.6 |
| 30 to <40 | 163 | 39.7 |
| 40 to <50 | 88 | 21.4 |
| ≥50 years | 26 | 6.3 |
| **Sex** | | |
| Male | 197 | 47.9 |
| Female | 214 | 52.1 |
| **Residence** | | |
| Urban residence | 245 | (59.6%), |
| Rural residence | 166 | 40.04% |
| **Marital Status** | | |
| Married | 210 | 51.1 |
| Never married | 72 | 17.5 |
| Divorced | 81 | 19.7 |
| Widowed | 48 | 19.7 |
| **Level of Educational status** | | |
| No education | 123 | 29.9 |
| Primary | 100 | 24.3 |
| Secondary | 115 | 28 |
| Tertiary and above | 73 | 17.8 |
| **Occupation** | | |
| Employed | 160 | 38.9 |
| Unemployed | 251 | 61.1 |

The highest anemia incidence density rate of adults living with HIV after enrolling HIV care was 74.58 per 100 person years of observation at 6-month follow-up period (95% CI = 0.24, 2.30) and decreased to 25.5/100 PYO (person years of observation) at 12-month follow-up period (95% CI = 0.11, 0.61).

The cumulative probability of anemia free survival of adults on HIV care at the median follow up period (4.09 year) was 79% (95% CI = 0.74, 0.83) and at the end of the fifth year was 71% (95% CI = .0.66, 0.76) (**Fig 1**).

Log rank (Mantel-Cox) test of equality of survival for categories of WHO clinical staging was done. The median anemia free survival time for patients with clinical stage III/IV was 53.33 months. Patients with stage I/II survive better and the difference was statistically significant between survival curves among the groups (p-value = 0.000) (Fig 2).

## Cox-regression analysis

Age category, sex, marital status, residence, educational status, occupation, disclosure status, Baseline WHO clinical stage, baseline CD4 cell count, functional status at enrolment, level of adherence, undernutrition, serum creatinine, serum ALT, past TB history, TB treatment, cotrimoxazole prophylaxis, isoniazid prophylaxis and recent viral load were eligible for multivariable analysis.

WHO clinical staging III/IV, undernutrition, increased serum creatinine and fair/poor adherence remained statistically significant predictors of increased anemia occurrence while increased ART duration was a significant predictor of decreased anemia occurrence (**Table 3**).

**Table 2. Baseline clinical and treatment related characteristics of adults on HIV care.**

| Variables | Number | Percent (%) |
|---|---|---|
| **Baseline CD4 count** | | |
| **> = 200 cells/ul** | 139 | 33.8 |
| **50 to<200 cell/ul** | 224 | 54.5 |
| **<50 cell/ul** | 48 | 11.7 |
| **WHO clinical stage** | | |
| **Clinical stage I/II** | 266 | 64.7 |
| **Clinical stage III/IV** | 145 | 35.3 |
| **BMI category** | | |
| **Normal** | 245 | 59.6 |
| **Obese** | 21 | 5.1 |
| **under nutrition** | 145 | 35.7 |
| **Serum creatinine** | | |
| **< = 1.1 IU/L** | 340 | 82.7 |
| **>1.1 IU/L** | 71 | 17.3 |
| **Serum alanine amino transferase** | | |
| **< = 50 IU/L** | 354 | 86.1 |
| **>50 IU/L** | 57 | 13.9 |
| **ART adherence** | | |
| **Good** | 331 | 80.5 |
| **fair/poor** | 80 | 19.5 |
| **Functional status** | | |
| **Working** | 298 | 72.5 |
| **Ambulatory** | 102 | 24.8 |
| **Bedridden** | 11 | 2.7 |
| **Past TB history** | | |
| **No** | 342 | 83.2 |
| **Yes** | 69 | 16.9 |
| **Cotrimoxazole prophylaxis** | | |
| **No** | 25 | 6.1 |
| **Yes** | 386 | 93.9 |
| **Isoniazid prophylaxis** | | |
| **No** | 82 | 20 |
| **Yes** | 329 | 80 |
| **Initial regimen** | | |
| **1j** | 9 | 2.2 |
| **1e** | 212 | 51.6 |
| **1a** | 50 | 12.2 |
| **ELSE**[*] | 64 | 15.6 |
| **1c** | 76 | 18.5 |
| **Changed Regimen** | | |
| **No** | 205 | 49.9 |
| **Yes** | 206 | 60.1 |
| **Changed regimen type** | | |
| **1j** | 99 | 48 |
| **1e** | 30 | 14.6 |
| **2f and 2h** | 26 | 12.6 |
| **ELSE**[**] | 26 | 12.6 |

*(Continued)*

**Table 2.** (Continued)

| Variables | Number | Percent (%) |
|---|---|---|
| 1c | 25 | 12.14 |
| **Recent viral load** | | |
| not detected | 384 | 93.4 |
| <150 cells/ul | 15 | 3.6 |
| 150 to <1000 | 5 | 1.2 |
| > = 1000 | 7 | 1.7 |

**ELSE*** = [1b(d4t-3TC-EFV), 1d(AZT-3TC-EFV), 1f(TDF+3TC-NVP), 1g(ABC+3TC-EFV) and 1h(ABC-3TC-NVP)], **ELSE**** = [1a(d4t-3TC-NVP), 1b(d4t-3TC-EFV), 1d(AZT-3TC-EFV), 1f(TDF+3TC-NVP), 1g(ABC+3TC-EFV) and 1h(ABC-3TC-NVP)], 1j = TDF-3TC-DTG, 2f = AZT-3TC-ATV/r, 2h = TDF-3TC-ATV/r.

## Discussion

This is a study of HIV positive adults whose age greater than or equal to 15 years old under DTGH HIV care in South Gondar zone, Northwest Ethiopia, to determine incidence of anemia and identify its predictors.

Almost one fifth (21.6%) of the study participants develop anemia giving an incidence density rate of 6.27(95% CI: 0.051, 0.077) per 100 person-years of observation. The overall incidence rate was lower than the study conducted in Northwest Ethiopia which was 27/100 PYO [41] and in the Capital city of Ethiopia which was 35.3/100 PYO [42]. This noticeable discrepancy might be related to early initiation of ART for participants in the current study irrespective of CD4 count and WHO clinical staging [35] and availability of new drug regimen (dolutegravir containing regimen). Dolutegravir drug has rapid viral suppression and higher genetic resistance to virus when compared with nonnucleoside reverse transcriptase inhibitors [40, 43, 44].

Similarly, the current study finding is also lower than the study conducted in Nigeria which was 38.2/100 PYO [23]. This variation might be related with utilization of zidovudine containing regimen in the previous study while this regimen has come down in the current study.

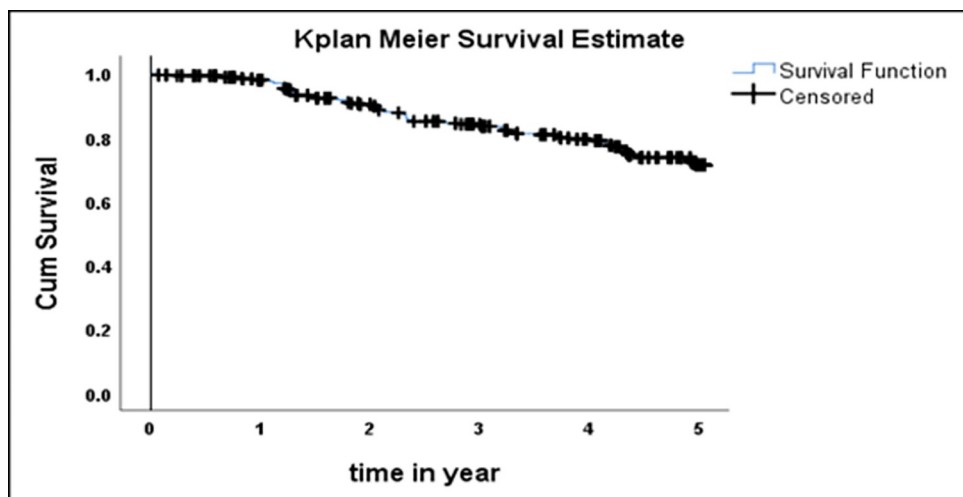

**Fig 1. Kaplan-Meier curve of anemia-free survival probability among HIV positive adults on ART at Debre Tabor Hospital, January 1, 2015 to December 30, 2019.**

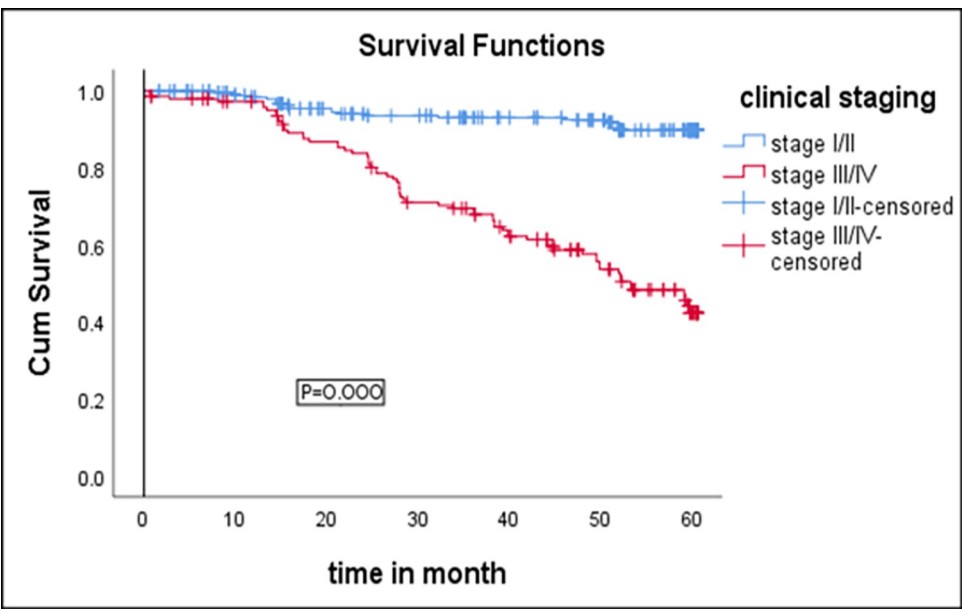

**Fig 2. Kaplan-Meier survival curve of anemia -free survival proportion based on WHO clinical staging among adults on HIV care.**

In the present study, adults living with HIV who were clinical stage III/IV at baseline have higher risk to develop anemia at any time as compared to those with WHO clinical stage I/II. This is supported by other evidence from Ethiopia [45], Tanzania [24] and South Africa [46]. This is due to the fact that, having advanced WHO clinical staging compromise immunity which leads to viral duplication and higher loads of opportunistic infections which results anemia via increased cytokine-mediated mylosuprression [47]. The current study suggests the need for anemia preventive measures along with HIV care for those patients who have advanced disease stage.

Additionally, early initiation of ART drug and good adherence should be encouraged to prevent the progression of advanced disease stage.

The current finding showed that, those HIV positive adult patients who were undernutrition at baseline have high risk to develop anemia at any time than those who were well-nourished. This finding is similar with study conducted in Northwest Ethiopia [41], Tanzania [24] and South Africa [46]. This is due to the fact that, patients who were undernutrition will have micronutrient deficiencies. The most common nutritional deficiencies are iron, folic acid, or vitamin B12 [29, 48].

This finding suggests that improving nutritional status of people living with HIV/AIDS taking ART drug through enhancing awareness of benefit of consuming balanced diet and micronutrient supplementation during follow-up may play significant role in decreasing anemia. Consuming a balanced diet helps the body in producing and proliferating enough amounts of red blood cells [49].

This finding also showed that, patients who have baseline serum creatinine level greater than 1.1 IU/L were more likely to develop anemia at any time as compared to those who have baseline serum creatinine less than or equal to 1.1 IU/L. Increased serum creatinine related to decrement of renal function to filter it which can result blunt erythropoietin production in response to lower hemoglobin concentration [50]. This finding suggests that, the need of erythropoietin treatment besides regular monitoring of hemoglobin concentration for patients who have elevated serum creatinine.

**Table 3. Predictors of anemia among HIV positive adults on ART at Debre Tabor General Hospital, January 1, 2015 to December 30, 2019.**

| Variables | | Censored | Anemia | Crude HR (95% CI) | P-value | Adjusted HR (95% CI) | P-value |
|---|---|---|---|---|---|---|---|
| Age category | 15 to <30 years | 106 | 28 | | | 1 | |
| | 30 to <40 years | 140 | 23 | 0.7 (0.41, 1.22) | 0.15 | 1.0(0.98, 1.02) | 0.729 |
| | 40 to <50 years | 65 | 23 | 1.37(0.79, 2.38) | 0.08 | 1.02(1.00, 1.04) | 0.063 |
| | > = 50 years | 11 | 15 | 3.37(1.8, 6.33) | 0.04 | 1.01(0.99, 1.03) | 0.434 |
| Sex | Male | 166 | 31 | | | 1 | |
| | Female | 156 | 58 | 1.61(1.04, 2.49) | 0.02 | 1.16(0.67, 2.02) | 0.593 |
| Marital status | Married | 180 | 30 | | | 1 | |
| | Single | 142 | 59 | 2.27(1.46, 3.52) | 0.01 | 1.53(0.9, 2.62) | 0.120 |
| Residence | Urban | 214 | 31 | | | 1 | |
| | Rural | 108 | 58 | 3.23(2.1, 5) | 0.03 | 0.9(0.51, 1.75) | 0.852 |
| Occupation | Employed | 142 | 18 | | | 1 | |
| | Unemployed | 180 | 71 | 2.76(1.64, 4.63) | 0.05 | 0.74(0.26, 2.13) | 0.582 |
| Educational status | secondary and above | 166 | 22 | | | 1 | |
| | Primary | 75 | 25 | 2.17(1.22, 3.85) | 0.02 | 2.11(0.87, 6.42) | 0.090 |
| | not educated | 81 | 42 | 3.24(1.93, 5.43) | 0.00 | 1.84(0.68, 4.98) | 0.230 |
| Disclosure status | Disclosed | 308 | 74 | | | 1 | |
| | not disclose | 14 | 15 | 4.21(2.4, 7.37) | 0.05 | 1.46(0.71,3.02) | 0.303 |
| Clinical staging | stage I/II | 245 | 21 | | | 1 | |
| | stage III/IV | 77 | 68 | 6.6(4.08, 10.87) | 0.00 | 1.03(1.01,1.06) | 0.001 * |
| CD4 count category | > = 200 cells/ul | 121 | 18 | | | 1 | |
| | 50 to <200 cells/ul | 173 | 51 | 1.74(1.02, 2.98) | 0.02 | 1.19(0.6, 2.34) | 0.618 |
| | <50 cells/ul | 28 | 20 | 3.23(1.71, 6.11) | 0.01 | 1.47(0.68,3.15) | 0.326 |
| BMI category | Normal | 231 | 14 | | | 1 | |
| | Overweight | 19 | 2 | 1.55(0.35, 6.83) | 0.18 | 1.7(0.32, 9) | 0.535 |
| | Undernutrition | 72 | 73 | 11.2(6.33,19.93) | 0.00 | 3.43(1.72, 6.83) | 0.000* |
| Serum creatinine category | < = 1.1 IU/L | 304 | 36 | | | 1 | |
| | >1.1 IU/L | 18 | 53 | 10.2(6.65,15.76) | 0.01 | 2.24(1.2, 4.2) | 0.011 * |
| Serum ALT category | < = 50 IU/L | 302 | 52 | | | 1 | |
| | >50 IU/L | 20 | 37 | 6.31(4.12, 9.67) | 0.03 | 1.49(0.85, 2.59) | 0.164 |
| Adherence category | Good | 294 | 37 | | | 1 | |
| | fair/poor | 28 | 52 | 8.11(5.23,12.47) | 0.01 | 2.9(1.43, 5.97) | 0.003* |
| Functional status | Working | 272 | 26 | | | 1 | |
| | Ambulatory | 47 | 55 | 6.88(4.31,10.98) | 0.08 | 1.04(0.51, 2.12) | 0.918 |
| | Bedridden | 3 | 8 | 11.8(5.33, 26.13 | 0.07 | 0.45(0.13, 1.55) | 0.206 |
| Past TB history | No | 286 | 56 | | | 1 | |
| | Yes | 36 | 33 | 3.62(2.36, 5.58) | 0.1 | 1.02(0.51, 2.03) | 0.953 |
| TB treatment | No | 269 | 48 | | | 1 | |
| | Yes | 53 | 41 | 3.48 (2.29, 5.29) | 0.06 | 2.00(0.91, 4.35) | 0.083 |
| Cotrimoxazole prophylaxis | No | 16 | 9 | | | 1 | |
| | Yes | 306 | 80 | 0.59(0.3,1.17) | 0.09 | 0.63(0.24,1.66) | 0.355 |
| Isoniazid prophylaxis | No | 44 | 38 | | | 1 | |
| | Yes | 278 | 51 | 0.27(0.17, 0.40) | 0.1 | 1.56(0.69,3.52) | 0.289 |
| Recent viral load | not detected | 307 | 77 | | | 1 | |
| | <150 cells/ul | 8 | 7 | 2.71(1.25, 5.88) | 0.13 | 1.38(0.57, 3.35) | 0.471 |
| | > = 150<1000 | 2 | 3 | 3.02(0.95, 9.57) | 0.08 | 0.29(0.07, 1.23) | 0.092 |
| | > = 1000 | 5 | 2 | 1.57(0.39, 6.4) | 0.18 | 2.02(0.43, 9.5) | 0.373 |

*(Continued)*

**Table 3.** (Continued)

| Variables | | Censored | Anemia | Crude HR (95% CI) | P-value | Adjusted HR (95% CI) | P-value |
|---|---|---|---|---|---|---|---|
| ART duration | ———— | ——— | —— | 0.99(0.98, 0.99) | | 0.98(0.97, 0.99) | 0.000* |

Marital status: Single includes Unmarried, widowed and divorced; BMI = body mass index; ALT = alanine aminotransferase; ART = anti-retroviral therapy; IU/L = international unit per liter; Statistical significance at 95% CI, P < 0.05

*reference statistically significant.

The present finding also showed that, patients who have fair/poor adherence were more likely to develop anemia at any time as compared to those who have good adherence. To the best of our understanding, patients who have missed their ART drug might be exposed to opportunistic infectious disease and increased the disease progression which leads to anemia via increased cytokine-mediated mylosuprression [47, 51]. This finding suggests that, more motivation, encouragement, and advice for HIV patients to adhere their ART drug therapy consistently so that to gain optimal therapeutic effect.

On the other hand, this study showed that, as ART duration increased the hazard of anemia was decreased by 2% times. It is in line with the study conducted in Tanzania [24]. Once ART has been initiated for HIV patients, there is quite a suppression of viral load that finally prevents and reverse anemia [27, 49, 52, 53] and as the duration of its utilization increased, the patients have sufficient time to recover from advanced disease progression [24]. Therefore, more Interventions to promote adherence to ART and continuous patient counseling are highly suggested.

## Limitation of the study

Secondary data which lacks some variables like food diversity and income status cannot be assessed.

## Conclusions

In sum, one out of five adult individuals develop anemia with a high incidence density rate.

Advanced clinical stage [III/IV], body mass index less than 18.5 Kg/m2, serum creatinine greater than 1.1 IU/L and fair/poor level of adherence were statistically significant predictors of anemia while increased ART duration decreases the risk of anemia. Even if the overall incidence rate of anemia was lower as compared to previous studies in Ethiopia, still the incidence of anemia was high. Therefore, Ethiopia Ministry of Health should design prevention measures by considering these factors besides of HIV care.

Early initiation of ART drug and good adherence level to ART should be encouraged to prevent advanced disease stage.

Well-integrated nutritional and HIV care system should be preserved to control the negative effect of undernutrition. Finally, a prospective study is highly suggested to include income status and dietary diversity assessment variables.

Anemia intervention measures would be designed for patients who have elevated serum creatinine along with HIV care.

Health care providers should give emphasizes adherence counseling, nutritional screening, renal function test and managements along with HIV care.

## Supporting information

**S1 File.**
(DOCX)

## Acknowledgments

The author's gratitude goes to Bahir Dar University, College of Medicine and Health Sciences for reviewing and giving ethical letter for this research. The authors would also like to extend their gratitude to Debre Tabor General Hospital, data collectors, and supervisor for their valuable contribution to the success of this study.

## Author Contributions

**Conceptualization:** Agimasie Tigabu, Yeshiwork Beyene, Temesgen Getaneh, Bogale Chekole, Tigist Gebremaryam, Ermias Sisay Chanie, Getachew Aragie, Getasew Legas, Getnet Dessie.

**Data curation:** Agimasie Tigabu, Yeshiwork Beyene, Temesgen Getaneh, Bogale Chekole, Tigist Gebremaryam, Ermias Sisay Chanie, Nigusie Selomom, Tamiru Alene, Getachew Aragie, Getasew Legas, Getnet Dessie.

**Formal analysis:** Agimasie Tigabu, Yeshiwork Beyene, Bogale Chekole, Tigist Gebremaryam, Ermias Sisay Chanie, Getachew Aragie, Getasew Legas, Getnet Dessie.

**Investigation:** Agimasie Tigabu, Ermias Sisay Chanie, Getnet Dessie.

**Methodology:** Agimasie Tigabu, Yeshiwork Beyene, Temesgen Getaneh, Tigist Gebremaryam, Ermias Sisay Chanie, Tamiru Alene, Getachew Aragie, Getasew Legas, Getnet Dessie.

**Project administration:** Agimasie Tigabu.

**Resources:** Agimasie Tigabu.

**Software:** Agimasie Tigabu, Tigist Gebremaryam, Getachew Aragie, Getasew Legas, Getnet Dessie.

**Supervision:** Agimasie Tigabu, Yeshiwork Beyene, Bogale Chekole, Tigist Gebremaryam, Ermias Sisay Chanie, Tamiru Alene, Getachew Aragie, Getasew Legas, Getnet Dessie.

**Validation:** Agimasie Tigabu, Yeshiwork Beyene, Temesgen Getaneh, Bogale Chekole, Ermias Sisay Chanie, Nigusie Selomom, Tamiru Alene, Getachew Aragie, Getnet Dessie.

**Visualization:** Agimasie Tigabu, Temesgen Getaneh, Bogale Chekole, Ermias Sisay Chanie, Nigusie Selomom, Getnet Dessie.

**Writing – original draft:** Agimasie Tigabu, Temesgen Getaneh, Ermias Sisay Chanie, Nigusie Selomom, Tamiru Alene, Getnet Dessie.

**Writing – review & editing:** Agimasie Tigabu, Yeshiwork Beyene, Temesgen Getaneh, Ermias Sisay Chanie, Nigusie Selomom, Getnet Dessie.

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
