## [Decision Letter · Decision Letter 0]

14 Jan 2021

PONE-D-20-26223

Incidence and Predictors of Anemia among Adults on HIV care at South  Gondar Zone Public General Hospital Northwest Ethiopia, 2020;  Retrospective Cohort Study.

PLOS ONE

Dear Dr. Tigabu,

Thank you for submitting your manuscript to PLOS ONE. After careful consideration, we feel that it has merit but does not fully meet PLOS ONE’s publication criteria as it currently stands. Therefore, we invite you to submit a revised version of the manuscript that addresses the points raised during the review process.

Please pay particular attention to ensuring you address each of the methodological concerns raised by the reviewers, and to clarifying details that will make the reporting of your study fully reproducible.

We look forward to receiving your revised manuscript.

Kind regards,

Jamie Males

Senior Editor

PLOS ONE

Journal Requirements:

2. In your ethics statement in the Methods section and in the online submission form, please provide additional information about the data used in your retrospective study. Specifically, please ensure that you have discussed whether all data were fully anonymized before you accessed them and/or whether the IRB or ethics committee waived the requirement for informed consent. If patients provided informed written consent to have data from their medical records used in research, please include this information.

3. Please include the date(s) on which you accessed the databases or records to obtain the data used in your study.

4. We noted in your submission details that a portion of your manuscript may have been presented or published elsewhere.

"the manuscript is processing for publication in BMC Infectious Disease"

Please clarify whether this publication was peer-reviewed and formally published. If this work was previously peer-reviewed and published, in the cover letter please provide the reason that this work does not constitute dual publication and should be included in the current manuscript.

5.  Thank you for stating the following in the Funding Section of your manuscript:

"This project was funded by Bahir Dar University. The funded grant was for data collection

 activities; namely, payments for data collectors, and the supervisors. However, the administrators

of Bahir Dar University had no role in study data analysis, data interpretation, or writing the

report. The corresponding author had full access to all data in the study and had final

responsibility for the decision to submit for publication."

"The author(s) received no specific funding for this work"

7. Please amend the manuscript submission data (via Edit Submission) to include authors Yeshiwork Beyene BSc, Temesgen Getaneh, Bogale Chekole, Tigist Gebremaryam, Tamiru Alene, Getenet Dessie.

8.  Your ethics statement should only appear in the Methods section of your manuscript. If your ethics statement is written in any section besides the Methods, please move it to the Methods section and delete it from any other section. Please ensure that your ethics statement is included in your manuscript, as the ethics statement entered into the online submission form will not be published alongside your manuscript.

Reviewers' comments:

Reviewer's Responses to Questions

**Comments to the Author**

1. Is the manuscript technically sound, and do the data support the conclusions?

Reviewer #1: Yes

Reviewer #2: Yes

2. Has the statistical analysis been performed appropriately and rigorously? 

Reviewer #1: Yes

Reviewer #2: I Don't Know

3. Have the authors made all data underlying the findings in their manuscript fully available?

Reviewer #1: No

Reviewer #2: Yes

4. Is the manuscript presented in an intelligible fashion and written in standard English?

Reviewer #1: Yes

Reviewer #2: Yes

5. Review Comments to the Author

Reviewer #1: General comments:

1. This institution-based retrospective cohort study conducted in Ethiopia will add value to the existing literature. In this study, they have found a high incidence rate if anemia, and body mass index, SCr, poor adherence level, and increased ART duration remained significant predictors for developing anemia. However, there is some study bias regarding study population pre-selection which needs to be analyzed and discussed in more detail. Moreover, language polishing is required throughout the text.

Major revisions

Introduction

1. It appeared to be exaggerated. It should be concise and problem-oriented. Please rewrite it and highlight the magnitude and burden of anemia in HIV in the context of Ethiopia and globally as well.

2. Please mention some evidence on mortality and morbidity rate secondary to anemia.

3. The line 58 t0 68 seems exaggerated. Please consider removing these lines.

4. Authors have mentioned the several causes of HIV-related anemia without proper citation of individual independent factors. I would suggest citing each risk factor accordingly. Add vital socio-demographic factors that have been potentially linked to anemia in HIV individuals.

5. Please state the clinical relevance of this study in Ethiopia.

Methods

1. In exclusion criteria, authors have stated only pregnant women and anemia at the beginning of the follow-up as exclusion criteria; however, failed to discuss other potential factors such as active gastric bleeding, and some drugs which may contribute to developing anemia during follow-up.

2. Please highlight the core outcome variables and covariates analyzed in the study.

3. For the sample size calculation, the authors have considered significant predictors for the incidence of anemia from the earlier study. However, it is not clear whether it is the incidence rate or predictors that the authors have considered. Please make it clear.

4. Data collection procedure: Please be consistent with the sentence while writing the conduct of the study. Please consider re-writing it (page no. 134-137).

5. Data were entered using EPI-data Version 3.1. Clear and completed were analyzed using

STATA Version 14 statistical software. The sentence is not clear. Please re-write it.

6. How the missing data were handled?

Results:

7. The median age of the entire cohort was 34 years (IQR; 12). More than half (52.1%)

157 of people living with HIV [PLHIV] were females and 323(78.6%) were Orthodox Christian

158 followed by Muslim 60(14.6%) and protestant 28(6.8%) follower. Majority, 245 (59.6%), of the159 patients came from urban areas. A total of 382 (92.9%) patients had disclosed their HIV status to160 either their husband/wife or other family members.

Please present the above figures in a table format.

8. Clinical and treatment related characteristics should be presented in table form

9. Do the patients have any underlying medical conditions? If yes, please present them in table

10. How about alcohol and smoking habit?

11. The predominant regiments initially prescribed were 174 combination of TDF, 3TC and EFV (1e) 212(51.6%) followed by AZT, 3TC and NVP (1c) 76(18.5%). Please consider changing the word regiments to regimens

12. Two hundred six (60.1%) patients had changed their initial regimen during the follow up period. From the total participants, 26(12.6%) patients switched to second-line HAART.

Some Anti-retroviral agents have a great impact on blood profile which may contribute to anemia. Please mention the type of regimens and its length that were switched to another regimen

13. How did the authors assess the ART adherence level? Please describe it in the methodology section.

14. Authors have described nutritional status based on BMI. I would rather describe BMI as normal, overweight and obese. Please change it.

15. What were the criteria for anemia? Please illustrate this in the methodology section

16. I would suggest highlighting the prevalence rate of anemia at enrollment in the study.

17. Please illustrate the baseline Hb levels of the enrolled patients.

18. SCr was categorized as: SCr <1.1 IU/L and >1IU/L. what are the clinical relevance of this classification.

19. What is the median ART duration? Please present it as a continuous variable as well as categorical ( e.g. <1 year vs >1 year)

20. Drug regimens may have the potential to influence anemia in this particular group of population. Therefore, I would advise authors to consider these analyses in both bivariate and multivariate analysis.

21. I would consider adding P-value for the Bivariate Cox-proportional hazard regression model

22. Please state the variables that were adjusted in the mulvariable Cox regression model.

Discussion

23. This noticeable discrepancy might be related to early initiation of ART for participants in the current study irrespective of CD4 count and WHO clinical staging (17) and availability of new drug regimen (dolutegravir containing regimen).

The above statement needs to be justified by the results shown in the study. Authors have failed to demonstrate the proportion of individuals who started early ART. The differences may be due to the availability of dolutegravir containing regimen, which need to be justified by existing literature.

24. Increased serum creatinine related to decrement of renal function to filter it which can result blunt erythropoietin production in response to lower hemoglobin concentration (30). This finding suggests that, the need of erythropoietin treatment beside regular monitoring of hemoglobin concentration for patients who have elevated serum

Based on the above statement, it is not clear whether patients have had some kind of renal impairment; and this should be clearly mentioned in inclusion/ exclusion criteria. Moreover, being SCr> 1.1 IU/L as a risk factor for developing anemia does not imply that the patients require erythropoietin treatment. More assessment with respect to renal function is needed.

25. This drug is associated with a more rapid viral suppression and higher genetic resistance when compared with nonnucleoside reverse transcriptase inhibitors.

The above statement seems irrelevant to correlate with the discrepancy seen in the study.

Conclusion

26. Please be focused on major findings and draw the conclusion accordingly. Furthermore, some suggestions seem to be exaggerated. Please consider re-writing it.

Reviewer #2: Incidence and Predictors of Anemia among Adults on HIV care at South Gondar Zone Public General Hospital Northwest Ethiopia, 2020; Retrospective Cohort Study.

General comments

• the manuscript needs minor language edits

• the research question lacks novelty; the authors have tried to assess if dolutegravir based regimen has effect on anemia incidence; however not convincegly assessed

• there are multiple studies addressing similar research question in Northwest Ethiopia

• the authors mentioned dolutegravir might decrease the risk of anemia; similarly they mentioned that it is introduced to Ethiopia at 2018 (a year or 2 before the end of the follow up); it will be great if the number of participants on dolutegravir based regimen is mentioned.

• the difference in the incidence of anemia in those with dolutegravir based regimen and others should have been mentioned with its statistical significance.

Specific comments

• the methodology requires more clarification

• the frequency of followup visits; the frequency of hemoglobin/hematocrit determination is not mentioned.

• the followup is mentioned to be 5 years , does this is apply for all the participants or the follow up varies for each patient

• Results

• The person year calculation, looks wrong, please look at it

• discussion

• the discussion shall focused and supported by your result findings, e.g

• erythropoeitin supplementation discussion is not supoorted by the result of this study

• References

• some of the references are not properly cited, e.g

• reference No 8; it looks there are 7 authors, however, there are only 3 authors.

6. PLOS authors have the option to publish the peer review history of their article (what does this mean?). If published, this will include your full peer review and any attached files.

Reviewer #1: **Yes: **Shiv Kumar Sah

Reviewer #2: **Yes: **Oumer Abdu Muhie

---

## [Author Response · Author response to Decision Letter 0]

7 Apr 2021

We have provided reviewers comment.

Senior Editor: 

Thank you for your comment. We have revised the manuscript and our file names to conform to PLOS’s style requirements.

2. In your ethics statement in the Methods section and in the online submission form, please provide additional information about the data used in your retrospective study

Thank you for pointing out this comment. We have added the ethical statements on page 6, lines 171 to 177.

3. Please include the date(s) on which you accessed the databases or records to obtain the data used in your study.

Thank you for the highlight. In the retrospective cohort study, the data collection period is similar to the study period. We have added the date on page 4, lines 111 to 112.

4. We noted in your submission details that a portion of your manuscript may have been presented or published elsewhere.

Thank you so much for your detailed observation. Currently, this manuscript has not been presented or published elsewhere.

5. Funding information should not appear in the Acknowledgments section or other areas of your manuscript.

Thank you for your constructive suggestion. We accept the comments and have agreed to your decision to take the funding information which presents in the Funding Statement section of the online submission form.________________________________________

6. Please ensure that you have an ORCID ID and that it is validated in Editorial Manager.

Thank you for your recommendation. I have authenticated the existing ORCID ID to the editorial manager.

7. Please amend the manuscript submission data (via Edit Submission) to include authors Yeshiwork Beyene BSc, Temesgen Getaneh, Bogale Chekole, Getasew Legas, Getachew Aragie, Tigist Gebremaryam, Tamiru Alene, Getenet Dessie.

Thank you so much for considering coauthors to include them in the online submitted manuscript. We have done.

Reviewers' comments: 

Reviewer #1:

Introduction

 It appeared to be exaggerated. It should be concise and problem-oriented. Please rewrite it and highlight the magnitude and burden of anemia in HIV in the context of Ethiopia and globally as well.

Thank you so much for your constructive comment. We have done it on page 3, line 73 to 76.________________________________________

 Please mention some evidence on mortality and morbidity rate secondary to anemia.

Thank you for your suggestion. Comorbidity was highlighted on page 3, line 77 to 78, and Mortality was highlighted on page 3, line 80 to 81. ________________________________________

 Authors have mentioned the several causes of HIV-related anemia without proper citation of individual independent factors. I would suggest citing each risk factor accordingly. Add vital socio-demographic factors that have been potentially linked to anemia in HIV individuals.

Thank you for pointing out this oversight. We have added vital socio-demographic factors that have been potentially linked to anemia in HIV individuals with the appropriate citation on page 3, line 87 to 90.________________________________________

 Please state the clinical relevance of this study in Ethiopia.

Thank you for this highlight. Conducting this study among adults on HIV care has a great significance to prevent, manage and reduce complications related to anemia for improving their health. 

It is also important to health professionals to encourage linkage of ART care to nutritional screening service, adherence counseling, and to policy makers to enhance decision making and planning of appropriate interventional strategies to prevent this comorbidity in Ethiopia.________________________________________

Methods

 In exclusion criteria, authors have stated only pregnant women and anemia at the beginning of the follow-up as exclusion criteria; however, failed to discuss other potential factors such as active gastric bleeding, and some drugs which may contribute to developing anemia during follow-up.

Thank you for this suggestion. Not only the mentioned potential factors but also we have assessed comorbid disorders that can lead to anemia like malaria, intestinal parasite, asthma, COPD, allergic disease but we did not get comorbid cases in our study.________________________________________

 Please highlight the core outcome variables and covariates analyzed in the study.

Thank you for pointing out the highlights. We have written the core outcome variable on page 9, line 198 to 203, and covariates used in the analysis showed on page 10 to 11, line 225 to 233.________________________________________

 For the sample size calculation, the authors have considered significant predictors for the incidence of anemia from the earlier study. However, it is not clear whether it is the incidence rate or predictors that the authors have considered. Please make it clear.

Thank you for this highlight. We have used significant predictors of anemia to calculate our sample size. We cannot calculate sample size by incidence rate by using single population proportion formula in survival analysis.________________________________________

 Data were entered using EPI-data Version 3.1. Clear and completed were analyzed using

STATA Version 14 statistical software. The sentence is not clear. Please re-write it.

Thank you for your detailed highlight. We have corrected it on page 6, line 158 to 159.________________________________________

 How the missing data were handled? 

Thank you for this highlight. We have removed variables that have greater than 5% missing by using the variable selection method.________________________________________

Results:

 The median age of the entire cohort was 34 years (IQR; 12). More than half (52.1%)

157 of people living with HIV [PLHIV] were females and 323(78.6%) were Orthodox Christian

158 followed by Muslim 60(14.6%) and protestant 28(6.8%) follower. Majority, 245 (59.6%), of the159 patients came from urban areas. A total of 382 (92.9%) patients had disclosed their HIV status to160 either their husband/wife or other family members.

Please present the above figures in a table format.

Thank you for this suggestion. We have presented the figures in table form on page 7, line 185 table 1.

 Clinical and treatment related characteristics should be presented in table form

Thank you for this suggestion. We have presented the figures in table form on page 8 to 9, line 193 table 2.________________________________________

 How about alcohol and smoking habit?

Thank you for this suggestion. These variables did not record in the patient’s chart since we have conducted a retrospective cohort study by chart review. ________________________________________

 Two hundred six (60.1%) patients had changed their initial regimen during the follow up period. From the total participants, 26(12.6%) patients switched to second-line HAART.

Some Anti-retroviral agents have a great impact on blood profile which may contribute to anemia. Please mention the type of regimens and its length that were switched to another regimen.

Thank you for this suggestion. As our study finding showed that, currently there is no regimen that cause to anemia rather they change their initial regimen due to new drug available (48%) and due to drug side effects (40.2%). On page 8, line 190 to 191.________________________________________

 How did the authors assess the ART adherence level? Please describe it in the methodology section.

Thank you for this suggestion. We have added the adherence assessment method in table form on page 5, line 131 to 133.________________________________________

 Authors have described nutritional status based on BMI. I would rather describe BMI as normal, overweight and obese. Please change it.

Thank you for this recommendation. Even if we have considered it, but we didn’t get data that is ≥ 30 kg/m2 to classify obese based on BMI classification. Rather we have gotten data which is BMI ≥ 25 kg/m2 < 30 kg/m2 and it is classified as overweight.

 What were the criteria for anemia? Please illustrate this in the methodology section

Thank you for this highlight. We have operationalized it on page 5, lines 126 to 127. ________________________________________

 I would suggest highlighting the prevalence rate of anemia at enrollment in the study.

Thank you for this suggestion. We didn’t assess the prevalence of anemia at enrolment since it is not our study’s objective. The first objective of our study is determining the incidence density rate by following participants for five years retrospectively who didn’t develop anemia at enrolment or who haven’t a previous record of anemia before the study starts. So that, the incidence density rate was 6.27/100 person-years of observation in our study. 

 Please illustrate the baseline Hemoglobin levels of the enrolled patients.

We have observed the level of serum hemoglobin of enrolled patients’ charts. The charts which have recorded normal serum hemoglobin at enrollment were entered into the study whereas charts which have recorded serum hemoglobin <13 g/dl before starting the study period were not included in the study as indicated from exclusion criteria on page 5, line 123 to 124.________________________________________

 SCr was categorized as: SCr <1.1 IU/L and >1 IU/L. what are the clinical relevance of this classification.

Thank you for this highlight. This classification has its own clinical relevance to assess kidney function. If the serum creatinine level is less than or equal to 1.1 IU/L, the renal function is normal to filter creatinine product. Whereas, increased serum creatinine level which is greater than 1.1 IU/L has related with a decrement of renal function to filter it which can result in blunt erythropoietin production in response to lower hemoglobin concentration.________________________________________

 What is the median ART duration? Please present it as a continuous variable as well as categorical ( e.g. <1 year vs >1 year)

Thank you so much for the suggestion. The mean of ART duration has stated on page 7, line 187. But we didn’t get a reference to make the variable categorical. 

 Drug regimens may have the potential to influence anemia in this particular group of population. Therefore, I would advise authors to consider these analyses in both bivariate and multivariate analysis.

Thank you so much for this suggestion. We have considered it but it is not fulfill the Cox proportional hazard regression model assumption checked by Schoenfeld residual and Log-Log plot tests. So that, we didn’t consider it for analysis.________________________________________

 I would consider adding P-value for the Bivariate Cox-proportional hazard regression model

Thank you for this suggestion. We have added P-value for Bivariable Cox- proportional hazard regression model on page 11 to 13, line 234, Table 3.________________________________________

 Please state the variables that were adjusted in the multivariable Cox regression model.

Thank you for your suggestion. We have stated the variables on page 11, line 236 to 231.________________________________________ Discussion

 This noticeable discrepancy might be related to early initiation of ART for participants in the current study irrespective of CD4 count and WHO clinical staging (17) and availability of new drug regimen (dolutegravir containing regimen).

The above statement needs to be justified by the results shown in the study. Authors have failed to demonstrate the proportion of individuals who started early ART. The differences may be due to the availability of dolutegravir containing regimen, which need to be justified by existing literature.

Thank you for this suggestion. We have justified it by existing literature on page 13, line 246 to 250. ________________________________________

 Increased serum creatinine related to decrement of renal function to filter it which can result blunt erythropoietin production in response to lower hemoglobin concentration (30). This finding suggests that, the need of erythropoietin treatment beside regular monitoring of hemoglobin concentration for patients who have elevated serum

Based on the above statement, it is not clear whether patients have had some kind of renal impairment; and this should be clearly mentioned in inclusion/ exclusion criteria. Moreover, being SCr> 1.1 IU/L as a risk factor for developing anemia does not imply that the patients require erythropoietin treatment. More assessment with respect to renal function is needed.

Thank you for this highlight. Increased serum creatinine does not mean renal impairment but it is an indication for reducing renal function. So that, erythropoietin treatment shall be given as prophylaxis or as a treatment for those who have increased serum creatinine level.________________________________________

Conclusion

 Please be focused on major findings and draw the conclusion accordingly. Furthermore, some suggestions seem to be exaggerated. Please consider re-writing it.

Thank you for the suggestion. We have taken the comment

---

## [Decision Letter · Decision Letter 1]

20 Sep 2021

PONE-D-20-26223R1Incidence and Predictors of Anemia among Adults on HIV care at South  Gondar Zone Public General Hospital Northwest Ethiopia, 2020;  Retrospective Cohort Study.PLOS ONE

Dear Mr Tigabu,

Thank you for submitting your manuscript to PLOS ONE. After careful consideration, we feel that it has merit but does not fully meet PLOS ONE’s publication criteria as it currently stands. Therefore, we invite you to submit a revised version of the manuscript that addresses the points raised during the review process.

We look forward to receiving your revised manuscript.

Kind regards,

Kwasi Torpey, MD PhD MPH

Academic Editor

PLOS ONE

Journal Requirements:

Additional Editor Comments (if provided):

I strongly suggest the authors to adequately address comments from Reviewer 1. Furthermore copyediting the final document to improve the language will be helpful prior to submission

Reviewers' comments:

Reviewer's Responses to Questions

**Comments to the Author**

1. If the authors have adequately addressed your comments raised in a previous round of review and you feel that this manuscript is now acceptable for publication, you may indicate that here to bypass the “Comments to the Author” section, enter your conflict of interest statement in the “Confidential to Editor” section, and submit your "Accept" recommendation.

Reviewer #1: (No Response)

Reviewer #2: All comments have been addressed

2. Is the manuscript technically sound, and do the data support the conclusions?

Reviewer #1: Partly

Reviewer #2: Yes

3. Has the statistical analysis been performed appropriately and rigorously? 

Reviewer #1: Yes

Reviewer #2: Yes

4. Have the authors made all data underlying the findings in their manuscript fully available?

Reviewer #1: Yes

Reviewer #2: Yes

5. Is the manuscript presented in an intelligible fashion and written in standard English?

Reviewer #1: No

Reviewer #2: Yes

6. Review Comments to the Author

Reviewer #1: Comments

The authors have attempted to address my previous concerns, yet there are many issues that need to be addressed properly before considering for publication. My specific comments are outlined below:

Abstract

-Anemia is a major public health problem worldwide which accounts 24.8% of the population. Few cross-sectional studies have been conducted on anemia and human immunodeficiency virus. However, it cannot address the incidence and predictors of anemia among human immunodeficiency virus -infected adults.

-The above statement does not support the relevance of the study, as there is a plethora of studies examining this issue globally as well as in Ethiopia also. Please consider rewriting the above statement again

-Moreover, the estimates presented in the background does not seem relevant at this stage

-Methods are not adequately described- such as population selection and outcome measures need to be clearly mentioned.

-The authors stated that a ethical clearance was obtained from the Institutional Review Board of Bahir Dar University, and also, we got permission letter to review charts from the concerned bodies in the hospital

-The above statement should be rephrased

-This is a similar confusion that which factors were from bivariate and multivariate analyses, and need to be very clearly described, and conclusion must be drawn from the appropriate findings.

Background:

-Line 63-67 seems irrelevant and exaggerated. It should be concise and problem-oriented.

-Please highlight the current burden of the disease globally, and especially in Ethiopia

-The authors failed to clearly summarize the current scenario of the topic. There seems to address the limitation of the current knowledge in this field, especially in Ethiopia

-The authors have revealed some data on the prevalence of anemia in Ethiopia; however, in this particular issue, providing data on the incidence of anemia in this particular setting is vital

-Line 87-89

The authors have indicated some contributing factors of developing anemia, yet there are many potential factors that contribute to this burden that needs to be considered.

-A strong rationale behind this study in this setting is needed

-Overall, the introduction needs to be fine-tuned to be more clear. The numerous grammatical errors and language needs to be considered.

Methods:

- In the exclusion criteria, authors have excluded only the pregnant women and anemia at the beginning of the follow-up . How about the other potential factors that may contribute to developing anemia during follow-up?

- Please highlight the core outcome variables and covariates analyzed in the study.

- Authors have assessed the adherence level based on the number of doses. I wonder how did the authors approach this, as the review was retrospective. Moreover, is this a standard tool for the assessment of the adherence level. Need citation and reliability of this tool.

- For the sample size calculation, the authors have used the significant predictors from previous studies. I wonder that there are numerous significant predictors for developing anemia, and which of them the author employed for calculating sample size.

-How the missing data were handled?

-In the methodology section, it is vital to include the clinical criteria for the diagnosis of anemia and its severity.

Results

-data should be presented in a well-fashioned manner. I see there are many results described which were not seen in table.

-The median age of the entire cohort was 34 years (IQR; 12). More than half (52.1%)

157 of people living with HIV [PLHIV] were females and 323(78.6%) were Orthodox Christian

158 followed by Muslim 60(14.6%) and protestant 28(6.8%) follower

The above statement should be presented in the table

Age group was categorized into :

15-<30;

30-<34

40-<50

≥50

Please state the rationale behind this classification. I do not see the age group of 35 to 30. Could you make clear about this?

-Clinical and treatment-related characteristics should be presented in table form

-Do the patients have any underlying medical conditions? If yes, please present them in table

- -it would be worth presenting baseline Hb level and prevalence of anemia.

- Some HAART regimens may contribute to developing anemia. Please consider this for analyses.

- Study participants were followed for a minimum of 0.67 and a maximum of 60.63 months.

-What does a minimum of 0.67 mean. Is this a good period for assessing the incidence of anemia?

- Please state the variables that were adjusted in the multivariable Cox regression model.

-This noticeable discrepancy might be related to early initiation of ART for participants in the current study irrespective of CD4 count and WHO clinical staging (17) and availability of new drug regimen (dolutegravir containing regimen).

The above statement needs to be justified by the results shown in the study. There is a need to describe the proportion of individuals who received ART, and need to be justified by the available evidence

-Increased serum creatinine related to decrement of renal function to filter it which can result blunt erythropoietin production in response to lower hemoglobin concentration (30). This finding suggests that, the need of erythropoietin treatment beside regular monitoring of hemoglobin concentration for patients who have elevated serum

-It is not clear whether patients have had some kind of renal impairment; and this should be clearly mentioned in inclusion/ exclusion criteria.

Reviewer #2: Thank you for addressing my concerns in the manuscript you submitted to the journal PLOS ONE. I appreciate your effort in that

7. PLOS authors have the option to publish the peer review history of their article (what does this mean?). If published, this will include your full peer review and any attached files.

Reviewer #1: **Yes: **Shiv Kumar Sah

Reviewer #2: **Yes: **Oumer Abdu Muhie

---

## [Author Response · Author response to Decision Letter 1]

11 Oct 2021

Agimasie Tigabu, BSc, MSc

Lecturer of Adult Health Nursing, Department of Nursing, 

College of Medicine and Health Science

Debre Tabor University, 

Debre Tabor, ETHIOPIA

Email: ethiomom23@gmail.com

Dr. Kwasi Torpey, MD PhD, MPH.

Aug 21 2020 2:32AM

My colleagues and I are happy to re-submit to you a revised version of our manuscript PONE-D- 20-26223R1, Incidence and Predictors of Anemia among Adults on HIV care: A Retrospective cohort study. For PLOS One, Thank you for giving us the opportunity to revise and resubmit this manuscript. I am resubmitting this revision on15th October 2021.

We thank the reviewers for their detailed and thoughtful suggestions. We have incorporated the recommended changes in the manuscript to the best of our ability and we feel that our revised manuscript is significantly improved as a result of your advice.

We have responded specifically to each suggestion in the memo below and have highlighted our changes in the manuscript text. In the memo, portions of the reviewers’ comments are reproduced in italic text, and our responses to the reviewers' comments are listed beneath in the Norman text.

The authors received no specific funding for this work.

Thank you once again for your time and attention to our manuscript. We look forward to working with you and the reviewers to move this manuscript closer to publication.

Best regards,

Agimasie Tigabu

REVIEWER COMMENTS AND RESPONSES

Reviewer's Responses to Questions

Comments to the Author 

1. If the authors have adequately addressed your comments raised in a previous round of review and you feel that this manuscript is now acceptable for publication, you may indicate that here to bypass the “Comments to the Author” section, enter your conflict of interest statement in the “Confidential to Editor” section, and submit your "Accept" recommendation.

Reviewer #1: (No Response)

Reviewer #2: All comments have been addressed 

Thank you for your suggestion. Actually, we didn’t get such comments if there will be, we will indicate our conflict of interest statement to bypass comments to Author section and enter it to Editor Section. After we got acceptance, we will submit the acceptance recommendation.

We thank you reviewer #2 for your detail understanding of the response for previous round of review comments. 

2. Is the manuscripts technically sound, and do the data support the conclusions?

Reviewer #1: Partly

Reviewer #2: Yes

Thank you so much for this oversight. As Reviewers said, the manuscript is a scientific research with the data that supports the conclusion and the conclusion has drawn based on the data presented. 

3. Has the statistical analysis been performed appropriately and rigorously?

Reviewer #1: Yes

Reviewer #2: Yes

Thank you so much for this suggestion. As the reviewers said, the statistical analysis has been performed appropriately and rigorously.________________________________________

4. Have the authors made all data underlying the findings in their manuscript fully available?

Reviewer #1: Yes

Reviewer #2: Yes

Thank you so much for this suggestion. All data underlying the findings are fully available within the manuscript.________________________________________

5. Is the manuscript presented in an intelligible fashion and written in Standard English?

Reviewer #1: No

Reviewer #2: Yes

Thank you so much for this suggestion. The manuscript is written in Standard English.________________________________________

6. PLOS authors have the option to publish the peer review history of their article (what does this mean?). If published, this will include your full peer review and any attached files.

Do you want your identity to be public for this peer review? For information about this choice, including consent withdrawal, please see our Privacy Policy.

Reviewer #1: Yes: Shiv Kumar Sah

Reviewer #2: Yes: Oumer Abdu Muhie

Thank you so much for this suggestion. We don’t want to our identity to be public. Thank you. 

Review Comments to the Author

Reviewer #1:

1. Abstract

Anemia is a major public health problem worldwide which accounts 24.8% of the population. Few cross-sectional studies have been conducted on anemia and human immunodeficiency virus. However, it cannot address the incidence and predictors of anemia among human immunodeficiency virus -infected adults.

-The above statement does not support the relevance of the study, as there is a plethora of studies examining this issue globally as well as in Ethiopia also. Please consider rewriting the above statement again

-Moreover, the estimates presented in the background does not seem relevant at this stage

Methods are not adequately described- such as population selection and outcome measures need to be clearly mentioned.

-The authors stated that a ethical clearance was obtained from the Institutional Review Board of Bahir Dar University, and also, we got permission letter to review charts from the concerned bodies in the hospital

-The above statement should be rephrased

-This is a similar confusion that which factors were from bivariate and multivariate analyses, and need to be very clearly described, and conclusion must be drawn from the appropriate findings.

Thank you so much for your constructive comment. We have rewritten and rephrased the statements per the given comments on page 2, line number 29 to 34. 

2. Background:

-Line 63-67 seems irrelevant and exaggerated. It should be concise and problem-oriented.

-Please highlight the current burden of the disease globally, and especially in Ethiopia

-The authors failed to clearly summarize the current scenario of the topic. There seems to address the limitation of the current knowledge in this field, especially in Ethiopia

-The authors have revealed some data on the prevalence of anemia in Ethiopia; however, in this particular issue, providing data on the incidence of anemia in this particular setting is vital

-Line 87-89

The authors have indicated some contributing factors of developing anemia, yet there are many potential factors that contribute to this burden that needs to be considered.

-A strong rationale behind this study in this setting is needed

-Overall, the introduction needs to be fine-tuned to be more clear. The numerous grammatical errors and language needs to be considered.

Thank you so much for your oversight. We have revised and edited the grammatical as well as language errors the background based on the given comments. 

3. Methods:

- In the exclusion criteria, authors have excluded only the pregnant women and anemia at the beginning of the follow-up. How about the other potential factors that may contribute to developing anemia during follow-up?

- Please highlight the core outcome variables and covariates analyzed in the study.

- Authors have assessed the adherence level based on the number of doses. I wonder how did the authors approach this, as the review was retrospective. Moreover, is this a standard tool for the assessment of the adherence level. Need citation and reliability of this tool.

- For the sample size calculation, the authors have used the significant predictors from previous studies. I wonder that there are numerous significant predictors for developing anemia, and which of them the author employed for calculating sample size.

-How the missing data were handled?

-In the methodology section, it is vital to include the clinical criteria for the diagnosis of anemia and its severity.

Thank you so much for your oversight. We have highlighted the core outcome variable in the operational definition part which was dichotomized into event and censored.

Adherence level: - missing dose is a standard tool for the assessment of the adherence level which is taken from ART intake form. We have stated it on page 5, line number 121 to 122.

On sample size calculation: - as you have mentioned that, we have used three significant predictors from previous studies. Those are nutritional status, past TB history and sex. We used the largest of all which is 434 by using nutritional status whereas by using past TB history and sex were given 190 and 44 respectively.

The missing data was handled by adding 10 % non-response rate and by excluding the study participants chart who have greater than or equal to 5% incompleteness.

4. Results

-data should be presented in a well-fashioned manner. I see there are many results described which were not seen in table.

-The median age of the entire cohort was 34 years (IQR; 12). More than half (52.1%)

157 of people living with HIV [PLHIV] were females and 323(78.6%) were Orthodox Christian

158 followed by Muslim 60(14.6%) and protestant 28(6.8%) follower

The above statement should be presented in the table

Age group was categorized into :

15-<30;

30-<34

40-<50

≥50

Please state the rationale behind this classification. I do not see the age group of 35 to 30. Could you make clear about this?

-Clinical and treatment-related characteristics should be presented in table form

-Do the patients have any underlying medical conditions? If yes, please present them in table

- -it would be worth presenting baseline Hb level and prevalence of anemia.

- Some HAART regimens may contribute to developing anemia. Please consider this for analyses.

- Study participants were followed for a minimum of 0.67 and a maximum of 60.63 months.

-What does a minimum of 0.67 mean? Is this a good period for assessing the incidence of anemia?

- Please state the variables that were adjusted in the multivariable Cox regression model. 

-Increased serum creatinine related to decrement of renal function to filter it which can result blunt erythropoietin production in response to lower hemoglobin concentration (30). This finding suggests that, the need of erythropoietin treatment besides regular monitoring of hemoglobin concentration for patients who have elevated serum

-It is not clear whether patients have had some kind of renal impairment; and this should be clearly mentioned in inclusion/ exclusion criteria.

Thank you so much for your constrictive comments. We described the frequencies of some variables in statement way and some are presented in table form.

Age categories: - we categorized it based on previous available literatures. 30 to 34 were written mistakenly. We have categorized it 30 years to 40 years on page 7. 

Prevalence of anemia is not the study objective rather we have determined anemia incidence density rate and identified its predictors including their survival probability. We have written it on page 9, line number 185 to 187. 

We have studied HAART regimens including changed regimen by considering a predictor variable for developing of anemia. In bivariable cox regression analysis, those regimens were not eligible for multivariable cox regression.

We have followed the study participants for five years duration. during the study period one study participant might be followed for a minimum of 0.67 months and one study participant might be followed for a maximum of 60.63 months.

 We have listed the variables which are adjusted in the multivariable Cox regression model on page 11, line number 215 to 217.

Based on the adjusted cox regression analysis result, study participants who have increased serum creatinine level have high risk to develop anemia as compared to who have normal serum creatinine level. Based on this finding, we have written implication for it. Whether the study participants have renal impairment or not if they have fulfilled the inclusion criteria are considered as the study participants. 

Reviewer #2:

Thank you for addressing my concerns in the manuscript you submitted to the journal PLOS ONE. I appreciate your effort in that.

Thank you so much.

---

## [Editor Report · Decision Letter 2]

19 Oct 2021

PONE-D-20-26223R2Incidence and Predictors of Anemia among Adults on HIV care at South  Gondar Zone Public General Hospital Northwest Ethiopia, 2020;  Retrospective Cohort Study.PLOS ONE

Dear Mr Agimasie Tigabu,

Thank you for submitting your manuscript to PLOS ONE. After careful consideration, we feel that it has merit but does not fully meet PLOS ONE’s publication criteria as it currently stands. Therefore, we invite you to submit a revised version of the manuscript that has been fully copyedited

We look forward to receiving your revised manuscript.

Kind regards,

Professor Kwasi Torpey, MD PhD MPH

Academic Editor

PLOS ONE

Journal Requirements:

Additional Editor Comments (if provided):

Thank you for the revisions. Comments have adequately been addressed. The manuscript requires significant copyediting. I suggest a fluent native speaker edits the manuscript prior to publication.
---

## [Author Response · Author response to Decision Letter 2]

29 Oct 2021

REVIEWER COMMENTS AND RESPONSES

Reviewer #1: 

Introduction

1. It appeared to be exaggerated. It should be concise and problem-oriented. Please rewrite it and highlight the magnitude and burden of anemia in HIV in the context of Ethiopia and globally as well. 

Thank you so much for your constructive comment. We have done it on page 3, line 69 to 70.

2. Please mention some evidence on mortality and morbidity rate secondary to anemia. 

Thank you for your suggestion. Comorbidity was highlighted on page 3, line 71 to 72, and Mortality was highlighted on page 3, line 74 to 76.

3. Authors have mentioned the several causes of HIV-related anemia without proper citation of individual independent factors. I would suggest citing each risk factor accordingly. Add vital socio-demographic factors that have been potentially linked to anemia in HIV individuals.

Thank you for pointing out this oversight. We have added vital socio-demographic factors that have been potentially linked to anemia in HIV individuals with the appropriate citation on page 3, line 80 to 82.

4. Please state the clinical relevance of this study in Ethiopia. 

Thank you for this highlight. Conducting this study among adults on HIV care has a great significance to prevent, manage and reduce complications related to anemia for improving their health. It is also important to health professionals to encourage linkage of ART care to nutritional screening service, adherence counseling, and to policy makers to enhance decision making and planning of appropriate interventional strategies to prevent this comorbidity in Ethiopia.

Methods

1. In exclusion criteria, authors have stated only pregnant women and anemia at the beginning of the follow-up as exclusion criteria; however, failed to discuss other potential factors such as active gastric bleeding, and some drugs which may contribute to developing anemia during follow-up.

Thank you for this suggestion. Not only the mentioned potential factors but also we have assessed comorbid disorders that can lead to anemia like malaria, intestinal parasite, asthma, COPD, allergic disease but we did not get comorbid cases in our study.

2. Please highlight the core outcome variables and covariates analyzed in the study. 

Thank you for pointing out the highlights. We have written the core outcome variable on page 9, line 187 to 189, and covariates used in the analysis showed on page 10 to 13, line 210 to 217 and on table 3.

3. For the sample size calculation, the authors have considered significant predictors for the incidence of anemia from the earlier study. However, it is not clear whether it is the incidence rate or predictors that the authors have considered. Please make it clear.

Thank you for this highlight. We have used significant predictors of anemia to calculate our sample size. We cannot calculate sample size by incidence rate by using single population proportion formula in survival analysis.

4. Data were entered using EPI-data Version 3.1. Clear and completed were analyzed using STATA Version 14 statistical software. The sentence is not clear. Please re-write it. 

Thank you for your detailed highlight. We have corrected it on page 6, line 147 to 148.

5. How the missing data were handled? Thank you for this highlight.

We have removed variables that have greater than 5% missing by using the variable selection method.

Results:

1. The median age of the entire cohort was 34 years (IQR; 12). More than half (52.1%) 157 of people living with HIV [PLHIV] were females and 323(78.6%) were Orthodox Christian 158 followed by Muslim 60(14.6%) and protestant 28(6.8%) follower. Majority, 245 (59.6%), of the159 patients came from urban areas. A total of 382 (92.9%) patients had disclosed their HIV status to160 either their husband/wife or other family members. Please present the above figures in a table format.

Thank you for this suggestion. We have presented the figures in table form on page 6, line 172 table 1.

2. Clinical and treatment related characteristics should be presented in table form Thank you for this suggestion.

We have presented the figures in table form on page 8 to 9, line 179 on table 2.

3. How about alcohol and smoking habit? 

Thank you for this suggestion. These variables did not record in the patient’s chart since we have conducted a retrospective cohort study by chart review.

4. Two hundred six (60.1%) patients had changed their initial regimen during the follow up period. From the total participants, 26(12.6%) patients switched to second-line HAART. Some Anti-retroviral agents have a great impact on blood profile which may contribute to anemia. Please mention the type of regimens and its length that were switched to another regimen. 

Thank you for this suggestion. As our study finding showed that, currently there is no regimen that cause to anemia rather they change their initial regimen due to new drug available (48%) and due to drug side effects (40.2%) on page 7, line 174 to 179.

5. How did the authors assess the ART adherence level? Please describe it in the methodology section. 

Thank you for this suggestion. We have added the adherence assessment method in table form on page 5, line 121 to 123.

6. Authors have described nutritional status based on BMI. I would rather describe BMI as normal, overweight and obese. Please change it.

Thank you for this recommendation. Even if we have considered it, but we didn’t get data that is ≥ 30 kg/m2 to classify obese based on BMI classification. Rather we have gotten data which is BMI ≥ 25 kg/m2 < 30 kg/m2 and it is classified as overweight.

7. What were the criteria for anemia? Please illustrate this in the methodology section 

Thank you for this highlight. We have operationalized it on page 4, lines 117 to 118.

8. I would suggest highlighting the prevalence rate of anemia at enrollment in the study.

Thank you for this suggestion. We didn’t assess the prevalence of anemia at enrolment since it is not our study’s objective. The first objective of our study is determining the incidence density rate by following participants for five years retrospectively who didn’t develop anemia at enrolment or who haven’t a previous record of anemia before the study starts. The incidence density rate of anemia was 6.27/100 person-years of observation in our study.

9. Please illustrate the baseline Hemoglobin levels of the enrolled patients. We have observed the level of serum hemoglobin of enrolled patients’ charts. The charts which have recorded normal serum hemoglobin at enrollment were entered into the study whereas charts which have recorded serum hemoglobin 1 IU/L. what are the clinical relevance of this classification.

Thank you for this highlight. This classification has its own clinical relevance to assess kidney function. If the serum creatinine level is less than or equal to 1.1 IU/L, the renal function is normal to filter creatinine product. Whereas, increased serum creatinine level which is greater than 1.1 IU/L has related with a decrement of renal function to filter it which can result in blunt erythropoietin production in response to lower hemoglobin concentration.

10. What is the median ART duration? Please present it as a continuous variable as well as categorical ( e.g. 1 year) 

Thank you so much for the suggestion. The mean of ART duration has stated on page 7, line 187. But we didn’t get a reference to make the variable categorical.

11. Drug regimens may have the potential to influence anemia in this particular group of population. Therefore, I would advise authors to consider these analyses in both bivariate and multivariate analysis. 

Thank you so much for this suggestion. We have considered it but it is not fulfill the Cox proportional hazard regression model assumption checked by Schoenfeld residual and Log-Log plot tests. Due to the above reason, we didn’t consider it for analysis.

I would consider adding P-value for the Bivariate Cox-proportional hazard regression model 

Thank you for this suggestion. We have added P-value for Bivariable Cox- proportional hazard regression model on page 11 to 13, line 218 to 219, Table 3.

Please state the variables that were adjusted in the multivariable Cox regression model. 

Thank you for your suggestion. We have stated the variables on page 11, line 215 to 217.

Discussion

1. This noticeable discrepancy might be related to early initiation of ART for participants in the current study irrespective of CD4 count and WHO clinical staging (17) and availability of new drug regimen (dolutegravir containing regimen) The above statement needs to be justified by the results shown in the study. Authors have failed to demonstrate the proportion of individuals who started early ART. The differences may be due to the availability of dolutegravir containing regimen, which need to be justified by existing literature.

Thank you for this suggestion. We have justified it by existing literature on page 13, line 233 to 234.

2. Increased serum creatinine related to decrement of renal function to filter it which can result blunt erythropoietin production in response to lower hemoglobin concentration (30). This finding suggests that, the need of erythropoietin treatment beside regular monitoring of hemoglobin concentration for patients who have elevated serum Based on the above statement, it is not clear whether patients have had some kind of renal impairment; and this should be clearly mentioned in inclusion/ exclusion criteria. Moreover, being SCr> 1.1 IU/L as a risk factor for developing anemia does not imply that the patients require erythropoietin treatment. More assessment with respect to renal function is needed. 

Thank you for this highlight. Increased serum creatinine does not mean renal impairment but it is an indication for reducing renal function. So that, erythropoietin treatment shall be given as prophylaxis or as a treatment for those who have increased serum creatinine level.

Conclusion

1. Please be focused on major findings and draw the conclusion accordingly. Furthermore, some suggestions seem to be exaggerated. Please consider re-writing it. 

Thank you for the suggestion. We have taken the comment

REVIEWER COMMENTS AND RESPONSES

Reviewer's Responses to Questions

Comments to the Author 

1. If the authors have adequately addressed your comments raised in a previous round of review and you feel that this manuscript is now acceptable for publication, you may indicate that here to bypass the “Comments to the Author” section, enter your conflict of interest statement in the “Confidential to Editor” section, and submit your "Accept" recommendation.

Reviewer #1: (No Response)

Reviewer #2: All comments have been addressed 

Thank you for your suggestion. Actually, we didn’t get such comments if there will be, we will indicate our conflict of interest statement to bypass comments to Author section and enter it to Editor Section. After we got acceptance, we will submit the acceptance recommendation.

We thank you reviewer #2 for your detail understanding of the response for previous round of review comments. 

2. Is the manuscripts technically sound, and do the data support the conclusions?

Reviewer #1: Partly

Reviewer #2: Yes

Thank you so much for this oversight. As Reviewers said, the manuscript is a scientific research with the data that supports the conclusion and the conclusion has drawn based on the data presented. 

3. Has the statistical analysis been performed appropriately and rigorously?

Reviewer #1: Yes

Reviewer #2: Yes

Thank you so much for this suggestion. As the reviewers said, the statistical analysis has been performed appropriately and rigorously.________________________________________

4. Have the authors made all data underlying the findings in their manuscript fully available?

Reviewer #1: Yes

Reviewer #2: Yes

Thank you so much for this suggestion. All data underlying the findings are fully available within the manuscript.________________________________________

5. Is the manuscript presented in an intelligible fashion and written in Standard English?

Reviewer #1: No

Reviewer #2: Yes

Thank you so much for this suggestion. The manuscript is written in Standard English.________________________________________

6. PLOS authors have the option to publish the peer review history of their article (what does this mean?). If published, this will include your full peer review and any attached files.

Do you want your identity to be public for this peer review? For information about this choice, including consent withdrawal, please see our Privacy Policy.

Reviewer #1: Yes: Shiv Kumar Sah

Reviewer #2: Yes: Oumer Abdu Muhie

Thank you so much for this suggestion. We don’t want to our identity to be public. Thank you. 

Review Comments to the Author

Reviewer #1:

1. Abstract

Anemia is a major public health problem worldwide which accounts 24.8% of the population. Few cross-sectional studies have been conducted on anemia and human immunodeficiency virus. However, it cannot address the incidence and predictors of anemia among human immunodeficiency virus -infected adults.

-The above statement does not support the relevance of the study, as there is a plethora of studies examining this issue globally as well as in Ethiopia also. Please consider rewriting the above statement again

-Moreover, the estimates presented in the background does not seem relevant at this stage

Methods are not adequately described- such as population selection and outcome measures need to be clearly mentioned.

-The authors stated that a ethical clearance was obtained from the Institutional Review Board of Bahir Dar University, and also, we got permission letter to review charts from the concerned bodies in the hospital

-The above statement should be rephrased

-This is a similar confusion that which factors were from bivariate and multivariate analyses, and need to be very clearly described, and conclusion must be drawn from the appropriate findings.

Thank you so much for your constructive comment. We have rewritten and rephrased the statements per the given comments on page 2, line number 29 to 34. 

2. Background:

-Line 63-67 seems irrelevant and exaggerated. It should be concise and problem-oriented.

-Please highlight the current burden of the disease globally, and especially in Ethiopia

-The authors failed to clearly summarize the current scenario of the topic. There seems to address the limitation of the current knowledge in this field, especially in Ethiopia

-The authors have revealed some data on the prevalence of anemia in Ethiopia; however, in this particular issue, providing data on the incidence of anemia in this particular setting is vital

-Line 87-89

The authors have indicated some contributing factors of developing anemia, yet there are many potential factors that contribute to this burden that needs to be considered.

-A strong rationale behind this study in this setting is needed

-Overall, the introduction needs to be fine-tuned to be more clear. The numerous grammatical errors and language needs to be considered.

Thank you so much for your oversight. We have revised and edited the grammatical as well as language errors of the background based on the given comments. 

3. Methods:

- In the exclusion criteria, authors have excluded only the pregnant women and anemia at the beginning of the follow-up. How about the other potential factors that may contribute to developing anemia during follow-up?

- Please highlight the core outcome variables and covariates analyzed in the study.

- Authors have assessed the adherence level based on the number of doses. I wonder how the authors approached this, as the review was retrospective. Moreover, is this a standard tool for the assessment of the adherence level? Need citation and reliability of this tool.

- For the sample size calculation, the authors have used the significant predictors from previous studies. I wonder that there are numerous significant predictors for developing anemia, and which of them the author employed for calculating sample size.

-How the missing data were handled?

-In the methodology section, it is vital to include the clinical criteria for the diagnosis of anemia and its severity.

Thank you so much for your oversight. We have highlighted the core outcome variable in the operational definition part which was dichotomized into event and censored.

Adherence level: - missing dose is a standard tool for the assessment of the adherence level which is taken from ART intake form. We have stated it on page 5, line number 121 to 122.

On sample size calculation: - as you have mentioned that, we have used three significant predictors from previous studies. Those are nutritional status, past TB history and sex. We used the largest of all which is 434 by using nutritional status whereas by using past TB history and sex were given 190 and 44 respectively.

The missing data was handled by adding 10 % non-response rate and by excluding the study participants chart who have greater than or equal to 5% incompleteness.

4. Results

-data should be presented in a well-fashioned manner. I see there are many results described which were not seen in table.

-The median age of the entire cohort was 34 years (IQR; 12). More than half (52.1%)

157 of people living with HIV [PLHIV] were females and 323(78.6%) were Orthodox Christian

158 followed by Muslim 60(14.6%) and protestant 28(6.8%) follower

The above statement should be presented in the table

Age group was categorized into:

15-<30;

30-<34

40-<50

≥50

Please state the rationale behind this classification. I do not see the age group of 35 to 30. Could you make clear about this?

-Clinical and treatment-related characteristics should be presented in table form

-Do the patients have any underlying medical conditions? If yes, please present them in table

- -it would be worth presenting baseline Hb level and prevalence of anemia.

- Some HAART regimens may contribute to developing anemia. Please consider this for analyses.

- Study participants were followed for a minimum of 0.67 and a maximum of 60.63 months.

-What does a minimum of 0.67 mean? Is this a good period for assessing the incidence of anemia?

- Please state the variables that were adjusted in the multivariable Cox regression model. 

-Increased serum creatinine related to decrement of renal function to filter it which can result blunt erythropoietin production in response to lower hemoglobin concentration (30). This finding suggests that, the need of erythropoietin treatment besides regular monitoring of hemoglobin concentration for patients who have elevated serum

-It is not clear whether patients have had some kind of renal impairment; and this should be clearly mentioned in inclusion/ exclusion criteria.

Thank you so much for your constrictive comments. We described the frequencies of some variables in statement way and some are presented in table form.

Age categories: - we categorized it based on previous available literatures. 30 to 34 were written mistakenly. We have categorized it 30 years to 40 years on page 7. 

Prevalence of anemia is not the study objective rather we have determined anemia incidence density rate and identified its predictors including their survival probability. We have written it on page 9, line number 185 to 189. 

We have studied HAART regimens including changed regimen by considering a predictor variable for developing of anemia. In bivariable cox regression analysis, those regimens were not eligible for multivariable cox regression.

We have followed the study participants for five years duration. during the study period one study participant might be followed for a minimum of 0.67 months and one study participant might be followed for a maximum of 60.63 months.

 We have listed the variables which are adjusted in the multivariable Cox regression model on page 11, line number 215 to 217.

Based on the adjusted cox regression analysis result, study participants who have increased serum creatinine level have high risk to develop anemia as compared to who have normal serum creatinine level. Based on this finding, we have written implication for it. Whether the study participants have renal impairment or not if they have fulfilled the inclusion criteria are considered as the study participants. 

Reviewer #2:

Thank you for addressing my concerns in the manuscript you submitted to the journal PLOS ONE. I appreciate your effort in that.

Thank you so much.

---

## [Editor Report · Decision Letter 3]

2 Nov 2021

Incidence and Predictors of Anemia among Adults on HIV care at South  Gondar Zone Public General Hospital Northwest Ethiopia, 2020;  Retrospective Cohort Study.

PONE-D-20-26223R3

Dear Mr Tigabu,

We’re pleased to inform you that your manuscript has been judged scientifically suitable for publication and will be formally accepted for publication once it meets all outstanding technical requirements.

Kind regards,

Professor Kwasi Torpey, MD PhD MPH

Academic Editor

PLOS ONE

Additional Editor Comments (optional):

Please correct reference No.53 and ensure it is consistent with journal guidelines
---

## [Editor Report · Acceptance letter]

2 Dec 2021

PONE-D-20-26223R3 

Incidence and Predictors of Anemia among Adults on HIV care at South  Gondar Zone Public General Hospital Northwest Ethiopia, 2020;  Retrospective Cohort Study. 

Dear Dr. Tigabu:

I'm pleased to inform you that your manuscript has been deemed suitable for publication in PLOS ONE. Congratulations! Your manuscript is now with our production department. 

Kind regards, 

on behalf of

Professor Kwasi Torpey 

Academic Editor

PLOS ONE